# Efficiently Troubleshooting Image Segmentation Models with Human-In-The-Loop

## Abstract

Image segmentation lays the foundation for many high-stakes vision applications such as autonomous driving and medical image analysis. It is, therefore, of great importance to not only improve the accuracy of segmentation models on well-established benchmarks, but also enhance their robustness in the real world so as to avoid sparse but fatal failures. In this paper, instead of chasing state-of-the-art performance on existing benchmarks, we turn our attention to a new challenging problem: how to efficiently expose failures of "top-performing" segmentation models in the real world and how to leverage such counterexamples to rectify the models. To achieve this with minimal human labelling effort, we first automatically sample a small set of images that are likely to falsify the target model from a large corpus of web images via the maximum discrepancy competition principle. We then propose a weakly labelling strategy to further reduce the number of false positives, before time-consuming pixel-level labelling by humans. Finally, we fine-tune the model to harness the identified failures, and repeat the whole process, resulting in an efficient and progressive framework for troubleshooting segmentation models. We demonstrate the feasibility of our framework using the semantic segmentation task in PASCAL VOC, and find that the fine-tuned model exhibits significantly improved generalization when applied to real-world images with greater content diversity. All experimental codes will be publicly released upon acceptance.

## 1 Introduction

Image segmentation (i.e., pixel-level image labelling) has recently risen to explosive popularity, due in part to its profound impact on many high-stakes vision applications, such as autonomous driving and medical image analysis. While the performance of segmentation models, as measured by excessively reused test sets (Everingham et al., 2010; Lin et al., 2014), keeps improving (Chen et al., 2018a; Badrinarayanan et al., 2017; Yu et al., 2018), two scientific questions have arisen to capture the community's curiosity, and motivate the current work:

**Q1:** *Do "top-performing" segmentation models on existing benchmarks generalize to the real world with much richer variations?*

**Q2:** *Can we identify and rectify the trained models' sparse but fatal mistakes, without incurring significant workload of human labelling?*

The answer to the first question is conceptually clearer, by taking reference to a series of recent work on image classification (Recht et al., 2019; Hendrycks et al., 2019). A typical test set for image classification can only include a maximum of ten thousands of images because human labelling (or verification of predicted labels) is expensive and time-consuming. Considering the high dimensionality of image space and the "human-level" performance of existing methods, such test sets may only spot an extremely small subset of possible mistakes that the model will make, suggesting their insufficiency to cover hard examples that may be encountered in the real world (Wang et al., 2020). The existence of natural adversarial examples (Hendrycks et al., 2019) also echos such hidden fragility of the classifiers to unseen examples, despite the impressive accuracy on existing benchmarks.

While the above problem has not been studied in the context of image segmentation, we argue that it would only be much *amplified* for two main reasons. First, segmentation benchmarks require

pixel-level dense annotation. Compared to classification databases, they are much more expensive, laborious, and error-prone to label[1], making existing segmentation datasets even more restricted in scale. Second, it is much harder for segmentation data to be class-balanced in the pixel level, making highly skewed class distributions notoriously common for this particular task (Kervadec et al., 2019; Bischke et al., 2018). Besides, the "universal" background class (often set to cover the distracting or uninteresting classes (Everingham et al., 2010)) adds additional complicacy to image segmentation (Mostajabi et al., 2015). Thus, it remains questionable to what extent the impressive performance on existing benchmarks can be interpreted as (or translated into) real-world robustness. If "top-performing" segmentation models make sparse yet catastrophic mistakes that have not been spotted beforehand, they will fall short of the need by high-stakes applications.

The answer to the second question constitutes the main body of our technical work. In order to identify sparse failures of existing segmentation models, it is necessary to expose them to a much larger corpus of real-world labelled images (on the order millions or even billions). This is, however, implausible due to the expensiveness of dense labelling in image segmentation. The core question essentially boils down to: how to efficiently decide what to label from the massive unlabelled images, such that a small number of annotated images maximally expose corner-case defects, and can be leveraged to improve the models.

In this paper, we introduce a two-stage framework with human-in-the-loop for efficiently troubleshooting image segmentation models (see Figure 1). The first stage automatically mines, from a large pool $\mathcal{D}$ of unlabelled images, a small image set $\mathcal{M}$, which are the most informative in exposing weaknesses of the target model. Specifically, inspired by previous studies on model falsification as model comparison (Wang &

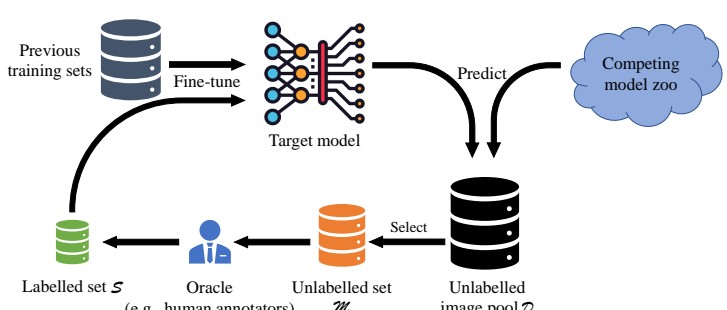

Figure 1: Proposed framework for troubleshooting segmentation models.

Simoncelli, 2008; Ma et al., 2018; Wang et al., 2020), we let the target model compete with a set of state-of-the-art methods with different design methodologies, and sample images by MAximizing the Discrepancy (MAD) between the methods. To reduce the number of false positives, we propose a weakly labelling method of filtering $\mathcal{M}$ to obtain a smaller refined set $\mathcal{S}$, subject to segmentation by human subjects. In the second stage, we fine-tune the target model to learn from the counterexamples in $\mathcal{S}$ without forgetting previously seen data. The two stages may be iterated, enabling progressive troubleshooting of image segmentation models. Experiments on PASCAL VOC (Everingham et al., 2010) demonstrate the feasibility of the proposed method to address this new challenging problem, where we successfully discover corner-case errors of a "top-performing" segmentation model (Chen et al., 2017), and fix it for improved generalization in the wild.

## 2    RELATED WORK

**MAD competition**    The proposed method takes inspiration from the MAD competition (Wang & Simoncelli, 2008; Wang et al., 2020) to efficiently spot model failures. Previous works focused on performance evaluation. We take one step further to also fix the model errors detected in the MAD competition. To the best of our knowledge, our work is the first to extend the MAD idea to image segmentation, where labeling efficiency is more desired since pixel-wise human annotation for image segmentation is much more time-consuming than image quality assessment (Wang & Simoncelli, 2008) and image classification (Wang et al., 2020) tasks previously explored.

---

[1]According to Everingham et al. (Everingham et al., 2010) and our practice, it can easily take ten times as long to segment an object than to draw a bounding box around it.

**Differential testing** Our method is also loosely related to the cross-disciplinary field of software/system testing, especially the differential testing technique (McKeeman, 1998). By providing the same tests to multiple software implementations, differential testing aims to find bug-exposing test samples that lead to different results. Programmers can then dig into these test cases for potential bug fixing. While debugging software is very different from troubleshooting machine learning algorithms, a handful of recent work explored this idea to find pitfalls of deep learning-based autonomous driving systems (Pei et al., 2017; Tian et al., 2018; Zhang et al., 2018). Aiming to be fully automated without human intervention, these methods have to make strong assumptions such as the ground truth labels can be determined by majority vote[2] or are unchanged under some synthetic image transformations[3] (e.g., brightness and contrast change, or style transfer). Therefore, it is unclear how to generalize the results obtained in such noisy and often unrealistic settings, to the real world with both great content fidelity and diversity.

**Adversarial examples** Introduced by Dalvi et al. (Dalvi et al., 2004) and reignited by Szegedy et al. (Szegedy et al., 2013), most adversarial attacks add small synthetic perturbations to inputs of computational models that cause them to make incorrect decisions. In image classification, Hendrycks et al. (Hendrycks et al., 2019) identified a set of natural images that behave like synthetic adversarial examples, which possess inherent transferability to falsify different image classifiers with the same type of errors. The selected counterexamples by the proposed framework might be treated as a new type of natural adversarial examples, that force the two models to make distinct predictions, therefore capable of fooling at least one model.

Similar as natural counterexamples focused in this work, synthetic adversarial examples pose security risks of deploying machine learning algorithms in real-world applications. A large body of research (Madry et al., 2018; Zhang et al., 2019) delves deep into adversarial training, trying to defend against adversarial perturbations at the expensive cost of sacrificing the generalization on original test sets without perturbations (Tsipras et al., 2019; Zhang et al., 2019; Schmidt et al., 2018). This seems to suggest a trade-off between generalization to real-world benign examples and robustness to adversarial attacks.

**Semantic segmentation with deep learning** Fully convolutional network (Long et al., 2015) was among the first deep architectures adopted for high-quality segmentation. Skip connection (Ronneberger et al., 2015), recurrent module (Zheng et al., 2015), max index pooling (Noh et al., 2015; Badrinarayanan et al., 2017), dilated convolution (Chen et al., 2014; Yu & Koltun, 2016; Chen et al., 2018a), and multi-scale training (Chen et al., 2016; 2018a) are typical strategies to boost the segmentation performance. Conditional random fields (Ladickỳ et al., 2010) used to dominate image segmentation before the advent of deep learning were also combined with convolutional networks to model spatial relationships (Zheng et al., 2015). We refer interested readers to (Minaee et al., 2020) for a comprehensive survey of this field.

## 3 PROPOSED METHOD

Suppose we have a target segmentation model $f_t : \mathbb{R}^{h \times w} \mapsto \{1, \cdots, c\}^{h \times w}$, where $h$ and $w$ are the height and width of the input image, and $c$ denotes the number of categories. Our goal is to efficiently identify and fix the failure cases of $f_t$ encountered in the real world, while minimizing human labelling effort in this process. We start by constructing a large image database $\mathcal{D}$, whose collection may be guided by the keywords that represent the $c$ categories. Rather than conducting large-scale subjective testing to obtain the ground truth segmentation map for each $x \in \mathcal{D}$, we choose to create a small subset of images $\mathcal{M} \subset \mathcal{D}$, which are strong candidates for revealing corner-case behaviors of $f_t$. To further reduce false positive examples in $\mathcal{M}$, we describe a method to gather a weak label for each $x \in M$ as an overall indicator of segmentation quality. Based on the labels, an even smaller set $\mathcal{S} \subset \mathcal{M}$ can be obtained for dense annotation by humans. Last, we fine-tune $f_t$

---

[2]As per (Hendrycks et al., 2019), machine learning algorithms with similar design philosophies tend to make common mistakes.

[3]In many areas of computer vision, methods trained on synthetic data cannot generalize to realistic data, and specialized techniques such as domain adaptation (Zhang et al., 2017; Zhao et al., 2019) have to be used to bridge the performance gap.

on the combination of $\mathcal{S}$ and previously trained data, in an attempt to learn from the found failures without forgetting (Li & Hoiem, 2017).

### 3.1 FAILURE IDENTIFICATION

**Constructing $\mathcal{M}$** Inspired by model falsification methodologies from computational vision (Wang & Simoncelli, 2008) and software engineering (McKeeman, 1998), we construct the set $\mathcal{M} = \{x_i\}_{i=1}^{n_2}$ by sampling the most "controversial" images from the large-scale unlabelled database $\mathcal{D} = \{x_i\}_{i=1}^{n_1}$, where $n_2 \ll n_1$. Specifically, given the target model $f_t$, we let it compete with a group of state-of-the-art segmentation models $\{g_j\}_{j=1}^m$ by maximizing the discrepancy (Wang et al., 2020) between $f_t$ and $g_j$ on $\mathcal{D}$:

$$\hat{x}^{(j)} = \underset{x \in \mathcal{D}}{\arg\max} \ d(f_t(x), g_j(x)), \quad j = 1, \dots, m, \tag{1}$$

where $d(\cdot)$ is a distance metric to gauge the dissimilarity between two segmentation maps (e.g., negative pixel accuracy, or mean region intersection over union (mIoU)). $\hat{x}^{(j)}$ represents the most controversial image according to $f_t$ and $g_j$, and therefore is the most informative in distinguishing between them. If the competing model $g_j$ performs at a high level, and differs from the target model $f_t$ in design, $\hat{x}^{(j)}$ is likely to be a failure of $f_t$.

To avoid identifying different instantiations of the same underlying root cause (Pei et al., 2017) and to encourage content diversity of the candidate images, we describe a "content-aware" method for constructing $\mathcal{M}$. We first partition $\mathcal{D}$ into $c$ overlapped subgroups $\{\mathcal{D}_{t,k}\}_{k=1}^c$ based on $f_t$'s predicted maps, where $x \in \mathcal{D}_{t,k}$ if at least one pixel in $f_t(x)$ belongs to the $k$-th category. After that, we add a content constraint by restricting the size of predicted pixels in the $k$-th category, i.e., $\sum \mathbb{1}[f_t(x) == k]/(h \times w)$, within the range of $[p_k, q_k]$. This allows excluding images of exceedingly large (or small) object sizes, which may be of less practical relevance. Moreover, instead of focusing on the most controversial example defined in Eq. (1), we look at top-$n_3$ images in $\mathcal{D}_{t,k}$ with $n_3$ largest distances computed by

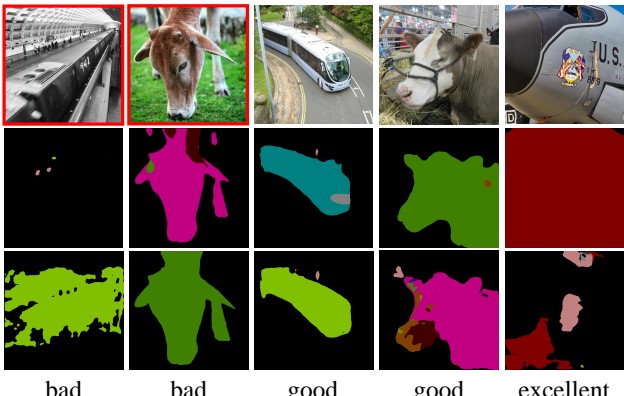

bad     bad     good     good     excellent

Figure 2: Purpose of weakly labelling. First row: Candidate images from $\mathcal{M}$. Counterexamples selected into $\mathcal{S}$ are highlighted with red rectangles, and the rest are false positives. Second and third rows: Predictions by the target and competing models, respectively. See Figure 4 for the color legend.

$$\{\hat{x}_i^{(j,k)}\}_{i=1}^{n_3} = \underset{\{x_i\}_{i=1}^{n_3} \in \mathcal{D}_{t,k}}{\arg\max} \ \sum_{i=1}^{n_3} d(f_t(x_i), g_j(x_i)), \quad j = 1, \dots, m, \ k = 1, \dots, c. \tag{2}$$

We then repeat this procedure, but with the roles of $f_t$ and $g_j$ reversed. That is, we partition $\mathcal{D}$ into $c$ subgroups $\{\mathcal{D}_{j,k}\}_{k=1}^c$ according to $g_j$, and solve the maximization problem over $\mathcal{D}_{j,k}$. Finally, we gather all candidate images to arrive at the set $\mathcal{M} = \{x_i\}_{i=1}^{n_2}$, where $n_2 \le 2mcn_3 \ll n_1$.[4]

**Constructing $\mathcal{S}$** Although images in the candidate set $\mathcal{M}$ have great potentials of being counterexamples of $f_t$, some false positives may be included (see Figure 2), especially when $g_j$ is inferior to $f_t$. In addition, no data screening is involved in the construction of $\mathcal{D}$, increasing chances of including images that are out-of-domain (e.g., falling out of the $c$ categories and/or containing inappropriate content). In view of these, we reduce false positives in $\mathcal{M}$ via a weakly labelling strategy. For each $x \in \mathcal{M}$, we ask human subjects to give a discrete score on an absolute category rating (ACR) scale to indicate the segmentation quality of $f_t(x)$. The labels on the scale are "bad", "poor", "fair", "good",

---

[4]We have $n_2 \le 2mcn_3$ because the same images may be optimal in different problems specified in Eq. (2).

and "excellent" (see Figure 8). We then rank all images in $\mathcal{M}$ by the mean opinion scores, and choose top-$n_4$ images with the smallest scores to form the counterexample set $\mathcal{S} = \{x_i\}_{i=1}^{n_4}$. Finally, we seek pixel-level segmentation results from human annotators for each image in $\mathcal{S}$ (see Figure 9).

## 3.2 MODEL RECTIFICATION

The labelled images in $\mathcal{S}$ give us a great opportunity to improve $f_t$ by learning from these failures. In order to be resistant to catastrophic forgetting (McCloskey & Cohen, 1989), the fine-tuning can also include labelled data previously used to train $f_t$. We may iterate through the whole procedure of failure identification and model rectification several rounds, leading to a progressive two-stage framework for efficiently troubleshooting segmentation models with human-in-the-loop.

When the iterative setting is enabled, the size of $\mathcal{S}$ is growing: $\mathcal{S} = \bigcup_{i=1}^{r} \mathcal{S}^{(i)}$, where $\mathcal{S}^{(i)}$ is the counterexample set created in the $i$-th fine-tuning round. Denoting the initial target model by $f_t^{(0)}$, we fine-tune $f_t^{(i-1)}$ on the combination of $\mathcal{S}$ accumulated in the previous $i - 1$ rounds and $\mathcal{S}^{(0)}$ used for pre-training. We summarize the proposed framework in Algorithm 1.

## 4 EXPERIMENTS

In this section, we use the semantic segmentation task defined in PASCAL VOC (Everingham et al., 2010) as a specific application to demonstrate the feasibility of our method. It is worth noting that the proposed framework can be applied to other segmentation tasks, such as those required in autonomous driving (Ess et al., 2009; Cordts et al., 2016) and medical image analysis (Ronneberger et al., 2015).

### 4.1 EXPERIMENTAL SETUPS

**Segmentation models** In our experiments, we choose the target model $f_t$ to be the state-of-the-art DeepLabV3Plus (Chen et al., 2018b) with DRN (Yu et al., 2017) as the

---

**Algorithm 1:** The proposed framework for efficiently troubleshooting segmentation models

**Input:** An unlabelled image set $\mathcal{D}$, a target model $f_t^{(0)}$ and the dataset $\mathcal{S}^{(0)}$ on which it is pre-trained, a group of competing models $\{g_j\}_{j=1}^{m}$, the maximum number $r$ of fine-tuning rounds, hyper-parameters $n_3, n_4$

**Output:** Improved $f_t^{(r)}$

1   $\mathcal{S} \leftarrow \emptyset$
2   **for** $j \leftarrow 1$ **to** $m$ **do**
3     Compute segmentation predictions $\{g_j(x), x \in \mathcal{D}\}$
4   **end**
5   **for** $i \leftarrow 0$ **to** $r - 1$ **do**
6     $\mathcal{M}^{(i+1)} \leftarrow \emptyset$
7     Compute segmentation predictions $\{f_t^{(i)}(x), x \in \mathcal{D}\}$
8     **for** $j \leftarrow 1$ **to** $m$ **do**
9      Compute the distances $\{d(f_t^{(i)}(x), g_j(x)), x \in \mathcal{D}\}$
10      Divide $\mathcal{D}$ into $c$ subgroups according to $f_t^{(i)}$
11      Filter images by the content constraint
12      Select top-$n_3$ images by solving Eq. (2) to form $\mathcal{M}_j$
13      $\mathcal{M}^{(i+1)} \leftarrow \mathcal{M}^{(i+1)} \bigcup \mathcal{M}_j$
14      Reverse the roles of $f_t^{(i)}$ and $g_j$, and repeat **Steps 10** to **13**
15     **end**
16     Source weak human scores for $\mathcal{M}^{(i+1)}$
17     Select top-$n_4$ images with the lowest quality scores and collect pixel-level labels from humans to form $\mathcal{S}^{(i+1)}$
18     $\mathcal{S} \leftarrow \mathcal{S} \bigcup \mathcal{S}^{(i+1)}$;
19     Fine-tune $f_t^{(i)}$ on the combination of $\mathcal{S}$ and $\mathcal{S}^{(0)}$
20   **end**

---

backbone (termed DeepLabV3P-DRN). We include five competing models: DeepLabV3Plus with ResNet101 (He et al., 2016) (termed DeepLabV3P-RN101), DeepLabV3 (Chen et al., 2017) with ResNet101 (termed DeepLabV3-RN101), DFN (Yu et al., 2018), Light-Weight RefineNet (Nekrasov et al., 2018) with ResNet50 and MobileNetV2 (Sandler et al., 2018) (termed LRefineNet-RN50 and LRefineNet-MNV2, respectively). Publicly available pre-trained weights on PASCAL VOC 2012 (Everingham et al., 2010) are used for all models. Following (Chen et al., 2017; 2018b), images are cropped and resized to $513 \times 513$ before inference.

**Constructing $\mathcal{D}$** We first briefly introduce the semantic segmentation database in PASCAL VOC (Everingham et al., 2010). It contains $1,464$ images for training (denoted by $\mathcal{S}^{(0)}$) and $1,149$ for validation (denoted by $\mathcal{T}^{(0)}$) with 20 scene categories (e.g., aeroplane, bicycle and person). We use the 20 class labels and combinations of them as keywords to crawl images from the Internet.

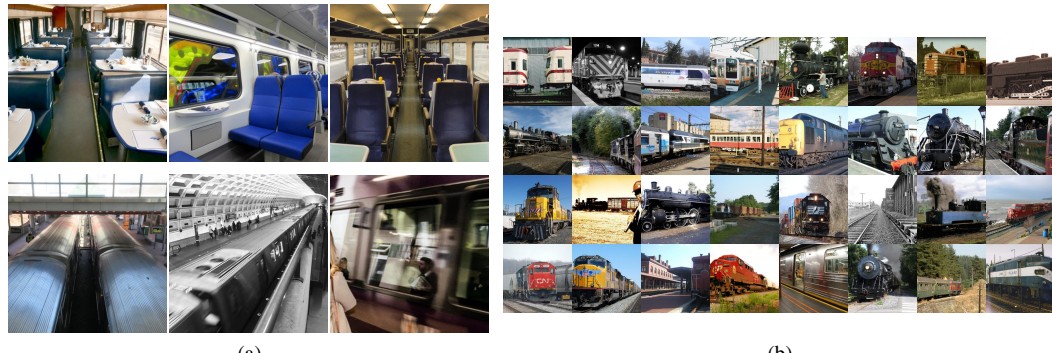

(a)         (b)

Figure 3: Visual comparison of images in (a) $\mathcal{S}$ and (b) PASCAL VOC validation set (denoted by $\mathcal{T}^{(0)}$).

Table 1: Segmentation results in terms of mIoU on both $\mathcal{T}^{(0)}$ and the unbiased test sets $\{\mathcal{T}^{(i)}\}_{i=1}^{r+1}$.

| Fine-tuning Round | | 0 | | 1 | | 2 | |
|---|---|---|---|---|---|---|---|
| Test Set | | $\mathcal{T}^{(0)}$ | $\mathcal{T}^{(1)}$ | $\mathcal{T}^{(0)}$ | $\mathcal{T}^{(2)}$ | $\mathcal{T}^{(0)}$ | $\mathcal{T}^{(3)}$ |
| Competing Models | DFN | 0.8037 | 0.1349 | 0.8037 | 0.1054 | 0.8037 | 0.1365 |
| | DeepLabV3-RN101 | 0.7795 | 0.1589 | 0.7795 | 0.1255 | 0.7795 | 0.1791 |
| | DeepLabV3P-RN101 | 0.7843 | 0.2555 | 0.7843 | 0.1978 | 0.7843 | 0.1483 |
| | LRefineNet-RN50 | 0.7710 | 0.1740 | 0.7710 | 0.1431 | 0.7710 | 0.2014 |
| | LRefineNet-MNV2 | 0.7125 | 0.1325 | 0.7125 | 0.1194 | 0.7125 | 0.1678 |
| Target Model | DeepLabV3P-DRN | 0.7887 | 0.1759 | 0.7827 | 0.2436 | 0.7828 | 0.4233 |

No other constraints are imposed during data collection. As a result, the database $\mathcal{D}$ includes a total of more than $40,000$ images, which is much larger than PASCAL VOC training and validation sets.

**Constructing $\mathcal{M}$**    We empirically set the two parameters $p_k$ and $q_k$ in the content constraint to the first and the third quartiles of the $k$-th category sizes of the training images in PASCAL VOC, respectively. To strike a good balance between the human effort of weakly labelling and the content diversity of $\mathcal{M}$, we choose $n_3 = 25$ in Eq. (2). Due to the existence of duplicated images, the actual sizes of $\{\mathcal{M}^{(i)}\}$ range from $1,500$ to $2,000$ for different rounds.

**Constructing $\mathcal{S}$**    We collect weak labels for images in $\mathcal{M}$ from three volunteer graduate students, who have adequate knowledge of computer vision, and are told the detailed purpose of the study. Each subject is asked to give an integer score between one and five for each image to represent one of the five categories, with a higher value indicating better segmentation quality. All out-of-domain images are given a score of positive infinity, meaning that any subject is granted to eliminate an image without agreement from the other two. The mean opinion scores averaged across all subjects are used to rank images in $\mathcal{M}$. The hyper-parameter $n_4$ in Algorithm 1 is set to 100. Representative examples in $\mathcal{S}$ are shown in Figure 3 and Figure 6 along with images in $\mathcal{T}^{(0)}$. It is clear that images in $\mathcal{S}$ are visually much harder.

We invite the same three students to provide ground truth segmentation results for images in $\mathcal{S}$, following the annotation guidance in PASCAL VOC, with the help of the online annotation tool Labelbox[5]. Our annotation process includes two stages with the goal of obtaining *consistent* segmentation maps. In the first stage, each subject is assigned one-third of the images to provide pixel-level labels. In the second stage, we carry out cross validation to improve the annotation consistency. Each student takes turn to check the segmentation maps completed by others, marking the positions and types of possible annotation errors. During cross checking, the subjects can discuss with each other to reach an agreement on a small part of ambiguous annotation.

**Iterative model correction**    We perform a total of $r = 2$ fine-tuning rounds. As suggested in (Chen et al., 2017), fine-tuning for each round is carried out by stochastic gradient descent with momentum

---

[5]https://labelbox.com/

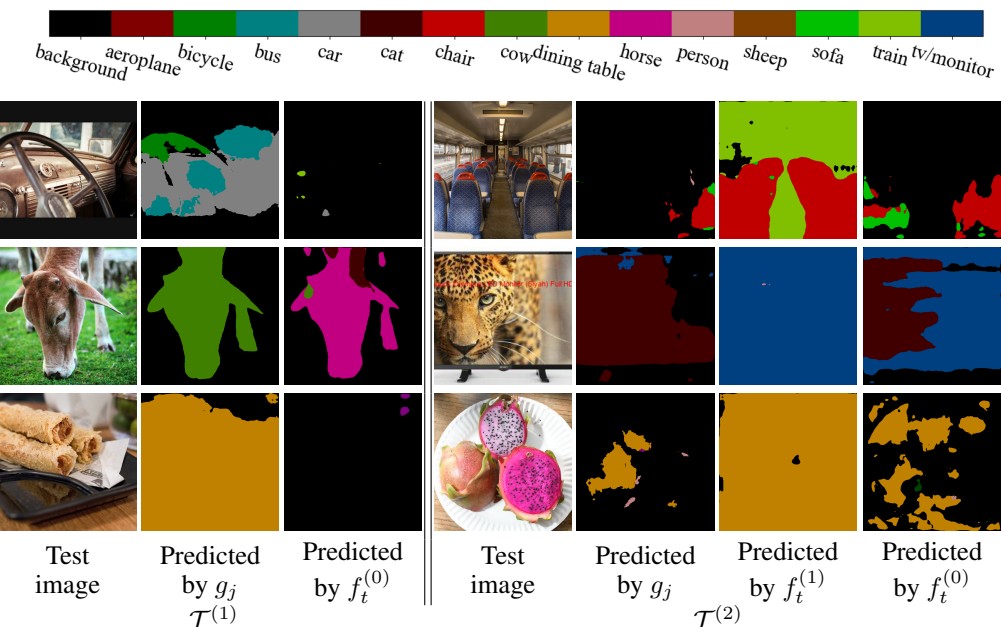

Figure 4: Left panel: Representative images in the test set $\mathcal{T}^{(1)}$ before fine-tuning the target model $f_t^{(0)}$. From the corresponding predicted segmentation maps, we find that the competing models $\{g_j\}$ successfully identify the failures of $f_t^{(0)}$. Right panel: Representative images in the test set $\mathcal{T}^{(2)}$ after the first round of fine-tuning, where we see that $f_t^{(1)}$ achieves noticeable performance improvements by learning from its failures in $\mathcal{S}^{(1)}$.

0.9 and weight decay $5 \times 10^{-4}$. We grid search the initial learning rate from $\{1, 5, 10\} \times 10^{-5}$, and choose the one with the highest mIoU on $\mathcal{T}^{(0)}$. The learning rate decay is guided by the polynomial policy. We set the mini-batch size and the maximum epoch number to 2 and 80, respectively.

**Model evaluation** How to reliably probe the generalization of computer vision models when deployed in the real world is by itself a challenging problem being actively investigated (Hendrycks et al., 2019; Arnab et al., 2018; Recht et al., 2019). Observing progress on $\mathcal{T}^{(0)}$ is not a wise choice because this set may only contain few catastrophic failures of the target model $f_t^{(i)}$. It may also be unfair to employ the counterexample set $\mathcal{S}^{(i+1)}$ to evaluate the relative progress of $f_t^{(i)}$ against the set of competing models $\{g_j\}_{j=1}^{m}$ due to the inclusion of the weakly labelling and filtering procedure. Inspired by the maximum discrepancy competition methodology for image classification (Wang et al., 2020), here we construct a new unbiased test set $\mathcal{T}^{(i+1)} \subset \mathcal{M}^{(i+1)}$ to compare $f_t^{(i)}$ with $\{g_j\}_{j=1}^{m}$ by adding the following constraints. First, similar as the construction of $\mathcal{S}$, all out-of-domain images are filtered out. Second, to encourage class diversity, we retain a maximum of four images that contain the same main foreground object (i.e., $\mathcal{T}^{(i+1)}$ has at most four "car" images). Third, to keep evaluation fair for the competing models, images used to fine-tune the target model $f_t^{(i)}$ are not included in $\mathcal{T}^{(i+1)}$. In our experiments, the size of $\mathcal{T}^{(i+1)}$ is set to 30.

### 4.2 MAIN RESULTS

**Quantitative results** We use a standard evaluation metric - mIoU to quantify the semantic segmentation performance. All results are listed in Table 1, where we find that, before the first round of fine-tuning, all models achieve competitive results on $\mathcal{T}^{(0)}$, implying close-to-saturation performance on PASCAL VOC. However, when tested on $\mathcal{T}^{(1)}$, the performance of all models drops significantly, indicating that many images in $\mathcal{T}^{(1)}$ are able to falsify both the target model $f_t^{(0)}$ and the associated competing model $g_j$. This also provides direct evidence that hard corner cases of existing segmentation models could be exposed. It is also proof-of-concept that the selection procedure is working

as intended. Moreover, the top-1 model on $\mathcal{T}^{(0)}$ does not necessarily perform the best on $\mathcal{T}^{(1)}$, conforming to the results in (Wang et al., 2020).

After the first round of fine-tuning, $f_t^{(1)}$ achieves noticeable improvements on $\mathcal{T}^{(2)}$, whereas all competing models experience different degrees of performance drops. This suggests that the target model begins to introspect and learn from its counterexamples in $\mathcal{S}^{(1)}$. After the second round of fine-tuning, the mIoU of $f_t^{(2)}$ on $\mathcal{T}^{(3)}$ is boosted by around $18\%$, surpassing all competing models by a larger margin than the previous round. This shows that our method successfully learns from and combines the best aspects of the competing models to fix its own defects, with approximately the same performance on $\mathcal{T}^{(0)}$. In our experiments, we only perform two rounds of fine-tuning due to limited computation and human resources, while we expect further performance gains under the proposed framework if tuned for more rounds.

**Qualitative results** We show representative test images before and after the first round of fine-tuning in Figure 4. Before fine-tuning, the competing models can effectively find defects of the target model with incorrect semantics and/or boundaries. After fine-tuning, the target model does a clearly better job on the corner-case samples. We also visually compare $f_t^{(0)}$ and $f_t^{(1)}$ on $\mathcal{T}^{(2)}$, where we observe the improved robustness to unseen corner-case images. Another interesting finding is that our method can fix a general class of model mistakes by learning from a small set of valuable failures. Figure 5 shows such an example, where the target model only sees 10 "car" images in $\mathcal{S}^{(1)}$, and is able to generalize to images with similar unusual viewpoints.

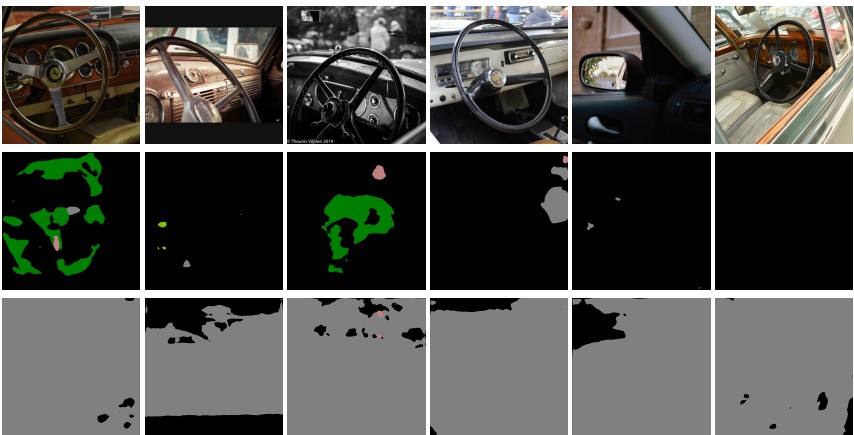

Figure 5: Representative "car" images from $\mathcal{M}^{(2)}$. First row: Test images. Second row: Predictions by $f_t^{(0)}$. Third row: Predictions by $f_t^{(1)}$ that is only exposed to a small set of "car" images in $\mathcal{S}^{(1)}$. The generalization of the target model on "car" images with unusual viewpoints is largely improved after the first round of fine-tuning. (All images shown in this figure are *not* in training set of $f_t^{(1)}$.)

## 5 CONCLUSION

We have formulated a new computer vision problem that aims to efficiently expose the failures of top-performing segmentation models and learn from such failures for improved generalization in the real world. We proposed an efficient human-in-the-loop framework for troubleshooting segmentation models, and demonstrated its promise under realistic settings. We hope this work sheds light on a new line of research that requires both machine vision and human interaction. In the future, we plan to explore the idea of leveraging natural failures for improving model robustness in broader vision problems, such as video understanding (Tran et al., 2015), computer-aided diagnosis (Mansoor et al., 2015), and computational neuroscience (Golan et al., 2019). Moreover, while the current work remains to focus on improving the model's standard generalization (with an emphasis on natural corner-case samples), our future study will investigate how this could be jointly optimized with adversarial robustness.

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

## A    COMPARISON WITH ENTROPY-BASED METHOD

Entropy has been used as an uncertainty metric to select hard samples (Joshi et al., 2009). The selected hard samples can be further used as training data to improve target model performance. In this section, we compare our method with the entropy-based method in (Joshi et al., 2009) to show the advantage of our method than traditional entropy-based methods. Since our goal is to improve model performance on open-world images instead of fixed benchmark datasets, we need to tell which method can achieve models with better performance on open-world images, which can be efficiently done by the MAD competition in (Wang et al., 2020).

For both methods, we conduct fine-tuning for one round and all experimental settings (including target model structure, human labelling budget, etc.) are kept identical. The only difference comes from the sampling strategy: our method samples according to Algorithm 1, while entropy-based method selects images with largest entropy. We then conduct MAD competition Wang et al. (2020) on the two models obtained from each method to compare their performance on open-world images. Numerical results are shown in Table 2. As we can see, the model achieved by our method is winning the MAD competition with a noticeable mIoU advantage, showing that the model achieved by our method has better performance on open-world images than the model fine-tuned by entropy-based method.

Table 2: MAD competition results on two models achieved by our method and entropy-base method respectively.

| Method | mIoU |
|---|---|
| Entropy-based | 19.65 |
| Ours | 26.53 |

## B    MORE VISUALIZATION RESULTS

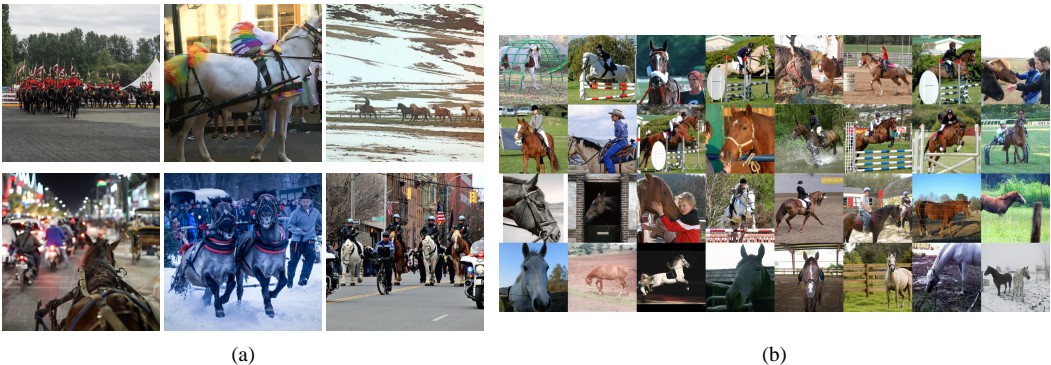

| (a) | (b) |

Figure 6: Visual comparison of images in (a) $\mathcal{S}$ and (b) PASCAL VOC validation set (denoted by $\mathcal{T}^{(0)}$).

## C    ABLATION STUDY

In order to show the testing results hold as we increase the size of $\mathcal{T}^{(i)}$ (denoted as $k$), we first evaluate the performance of the six models with different number of testing images, and conduct a global ranking among the six models. We use top-$k$ ranking to denote the global ranking evaluated with $k$ testing images. Then we calculate the SRCC values between the top-40 ranking (as reference) and other top-$k$ rankings with $k \in \{20, 21, \cdots, 39\}$. As shown in Figure 7, the ranking results remain stable when $k > 25$. And our active finetuned target model always achieves the best performance on $\mathcal{T}^{(1)}$ among the six models, for any $k$ between 20 and 40.

## D    SUBJECTIVE TESTING GUIS

Graphical user interfaces (GUIs) for our weakly and pixel-wise labelling experiments are shown in Figure 8 and Figure 9, respectively. For weakly labelling experiments, we build our own GUI: an

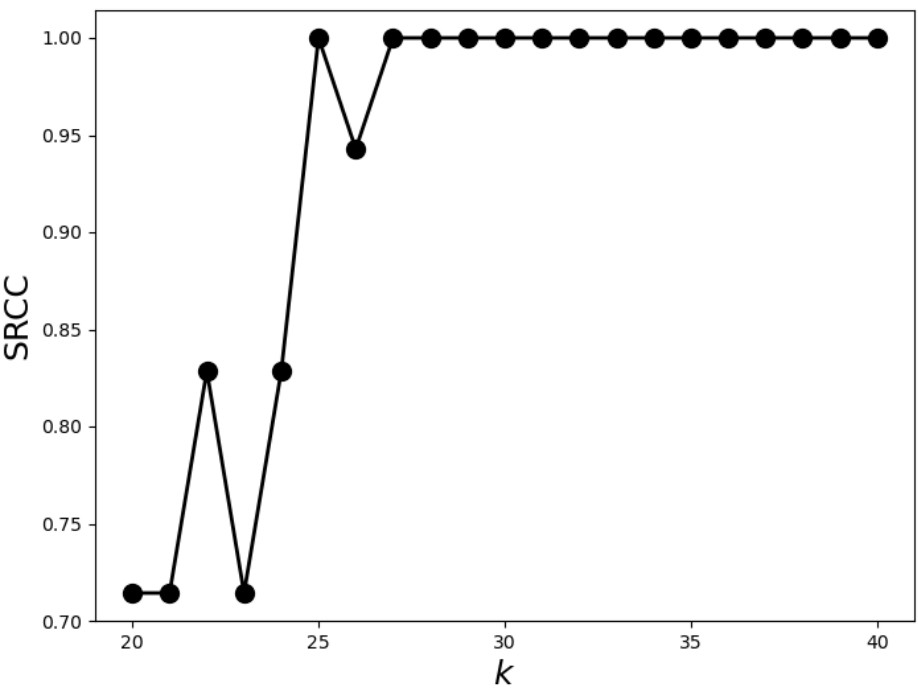

Figure 7: The SRCC values between the top-$40$ ranking and other top-$k$ rankings.

image is rendered alongside the prediction made by the target model; a scale-and-slider applet is utilized to collect the absolute category rating score of that image as described in Section 3.1. For pixel-wise segmentation labelling experiments, we use LabelBox segmentation template[6] to build our GUI.

---

[6]https://labelbox.com/product/image-segmentation

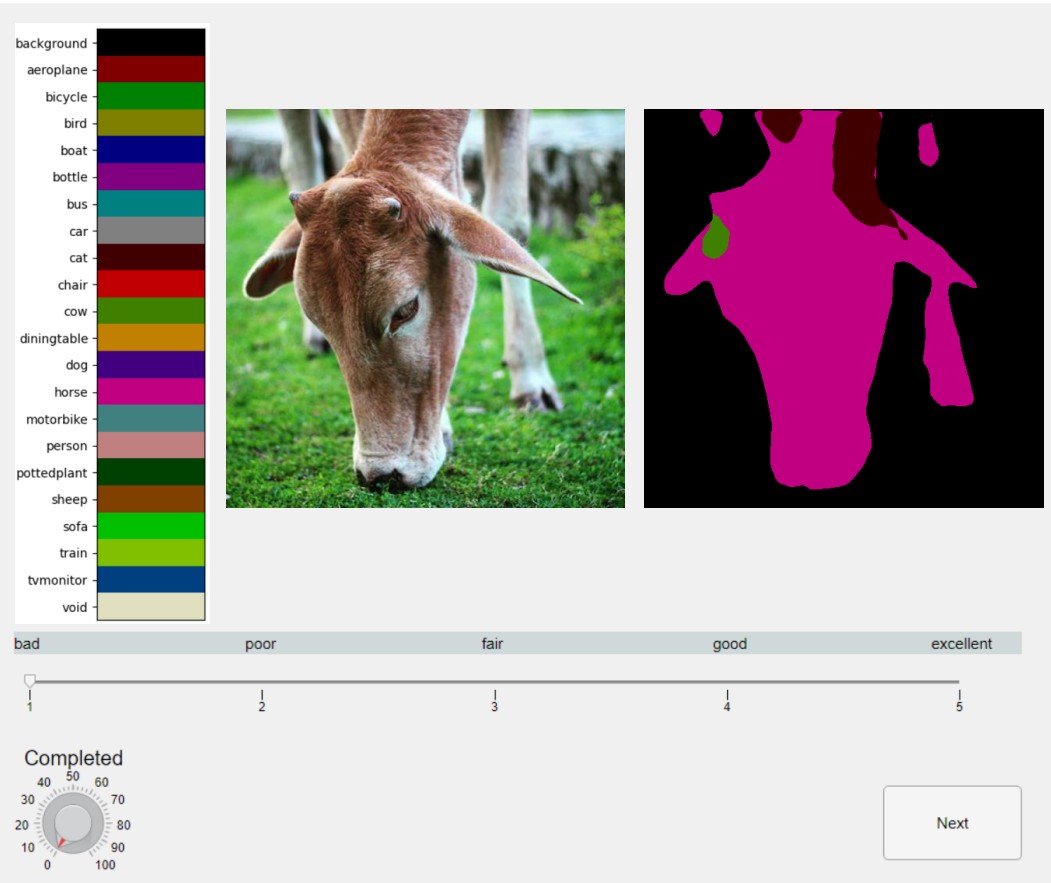

Figure 8: GUI for our weakly labelling experiments.

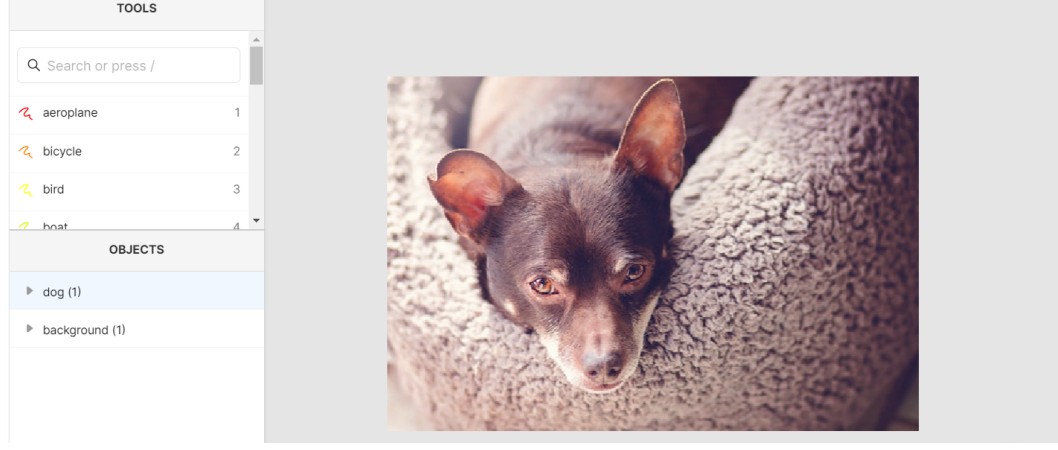

Figure 9: LabelBox GUI for our pixel-level segmentation labelling experiments.

