# OpenReview forum: "Efficiently Troubleshooting Image Segmentation Models with Human-In-The-Loop"
_ICLR.cc/2021/Conference — Reject_

### Official Review · AnonReviewer1 · 2020-10-27

**Rating:** 8
**Confidence:** 4

**Review:**

This work used a variety of existing segmentation algorithms to discover most "controversial" samples from massive online unlabeled images. Those representative controversial samples were believed to have the best chance to confuse the algorithm being trained and to expose its weakness. They are rated by annotators on a spectrum from bad to excellent, and segmentation masks are collected from human annotators for the “worst” images. Several clever measures were taken to reduce human labor.

The paper addressed an important and somewhat overlooked problem for segmentation and deep learning in the general. Leveraging those counterexamples to improve the segmentation models' generalization performance on unseen images seems to be novel in this field. This looks like a special case of finding natural adversarial examples from unlabeled data. The proposed solution is intuitive and logical, with lots of practical considerations made for its feasibility. The manuscript is also very well written, and the literature review is especially comprehensive.

This work extends from MAD, ICLR 2020 (https://openreview.net/forum?id=rJehNT4YPr); but it also presents with two nontrivial and interesting innovations: (1) generalizing it to a dense prediction task, which requires revising the human labelling strategy in subjective experiments. Weakly-supervised labeling is more practical for segmentation; and (2) extending to active training/tuning, leveraging the selected hard examples to improve the segmentation model for multiple rounds. The tuned models were shown to improve their robustness remarkably on spotted catastrophic mistakes, while preserving their performance on canonical testing sets.

This approach is also an instance of the basic active learning paradigm that iterates between spotting hard examples, labeling them, and tuning the model. The method of finding hard examples by competing with a human oracle seems to be novel though.

I noticed in the Appendix, the authors also compared their method with entropy-based active learning – a vanilla baseline needing no competing model. Their model seems to achieve a good mIoU advantage, showing the new proposed way has better performance on open-world images than simple entropy-based fine-tuning. That is perhaps understandable, since the proposed new method also costs more human inspection efforts. However, I’d be interested to see the authors to compare the hard examples selected by their method and by entropy-based one: are there any distinct pattern or notable trend? Is there anything that the entropy-based method can clear miss but your method can pick up?

Besides, this paper uses different deep networks trained as competing models. Why wouldn’t those models more tend to make similar mistakes, due to same training dataset and model type? What if using learning-based but non-deep segmentation models to compete? What if using segmentation models trained on other datasets to compete?

---

> ### Author Response · Authors · 2020-11-20
> **Response to Reviewer1**
>
> Thank you for your nice, insightful, and constructive comments!
>
> Q: Difference between images selected by our method and entropy-based method
>
> A: Compared with those selected by the entropy-based method, images selected by our method tend to be more diverse, thus covering more different failure cases, which is beneficial for improving the target model performance in the finetune process.
>
> Q: Use other methods as competing models
>
> A: Thank you for raising this excellent question. Adopting MAD competition has two prerequisites that one shall balance: (1) the two models are different behaviors, so they more easily “agree” on controversial examples; (2) both models are comparably strong performers, because only in this way we can assume that, if a controversial example is found, it is highly likely to be a corner case falsifying one of the two classifiers (i.e., Case II in MAD paper).
>
> In our work, the second point is especially important for us. We aim to troubleshoot an already strong segmentation model using another competing model. If the competing model is substantially weaker (even diverse), then when the two models disagree, most of the spotted controversial examples would turn out to be errors made by the weak competitor. That cannot achieve our goal to spot the strong model’s (presumably very sparse) errors and improve it.
>
> That explains why we do not choose non-deep segmentation models or models trained on other datasets: they will have too “weak” performances compared to a SOTA deep segmentation model trained on this specific dataset, and therefore cannot meet our goal. Instead, we find that other deep segmentation models (exploiting different modules or backbones) can make effective competitors that balances inter-model diversity and each model’s strength.

---

### Official Review · AnonReviewer2 · 2020-10-28
**The work promises more than it delivers. Approach could be useful, however, experiments and discussion of prior work appears to be lacking.**

**Rating:** 3
**Confidence:** 4

**Review:**

Annotating images for training of segmentation models is time consuming and it can be difficult to annotate enough examples to ensure good performance on the rare difficult examples that often occur when methods are applied to real world data. These cases are referred to as corner-cases. The paper therefore proposes a measure based on the discrepancy of a group of segmentation models to identify more valuable images to annotate and add to the training data in a iterative fashion. The approach is tested on the PASCAL VOC database.

The problem is important and relevant. I find the motivation clear, however, I disagree somewhat with the reasoning in parts of the introduction (see detailed comments). The approach is reasonable, and there is some evidence that performance is improved on these specific corner-cases by the addition of similarly identified corner-cases to the training set (Table 1). I am not sure how the approach would generalize, however. As I understand it the test-sets where performance is improved consists entirely of corner-cases identified in a similar manner, so whether the approach will generalize, depends entirely on how representable corner-cases identified in this manner are. It would have been much better to use a independent dataset of corner-cases identified manually.


Detailed comments

- "First, segmentation benchmarks require pixel-level dense annotation", I do not believe this is necessarily true, and there is little need to state this. One could certainly think of useful benchmarks with hard examples only. There are also examples of benchmarks where the groundtruth consists of computer segmentations corrected by humans.

- "Second, it is much harder for segmentation data to be class-balanced in the pixel level, making highly skewed class distributions notoriously common for this particular task", "Besides, the “universal” background class (often set to cover the distracting or uninteresting classes (Everingham et al., 2010)) adds additional complicacy to image segmentation (Mostajabi et al., 2015).", while this may be true for training datasets, I do not see how this is a problem for benchmarks necessarily.

- I find the description of the construction of the test dataset used in the different iterations unclear. It is my understanding, but I am not actually sure so it would be good to have the approach clarified, that the test datasets ($T^{(1)}$, $T^{(2)}$, and $T^{(3)}$) of iteration 1, 2, and 3 are hard examples, and are thus biased towards the methods involved. That is, they consist of examples that the proposed segmentation model disagrees with the "competing" models the most on. It is clear from the Table that these images are selected both based on mistakes of the competing models and mistakes of the target model. After one and two iterations we see that the target model now does much better on the next iteration of hard examples, but we really do not know how representative these hard examples are. If the methods tend to disagree on a limit number of typical cases, then these cases will be added to the training set and it is not so surprising that improvements in the target model is seen. To evaluate how this approach generalizes to the real world, an independent dataset would have to be used.

- "This also provides direct evidence that existing segmentation models could be particularly weak at certain real-world generalization, which is not surprising because the 1,464 training images are deemed to be extremely sparsely distributed in the space of natural images." I am not sure I agree with the evidence part. The dataset in question is selected to be hard (as far as I understand), so it is not surprising that the methods perform worse on it and does not say much about generalization. Essentially this is just evidence that the selection procedure is working as intended.

Clarity
While I believe I understand most of the manuscript to an acceptable level, it contains quite a lot of sentences that would benefit from some editting. Some examples below.

- Introduction is wordy, with long and difficult to understand sentences. Words that mean different things than the authors probably intended are also used, see examples "boosting", "spot", "cover" and wrong words are often used.

- "While the performance of segmentation models, as measured by excessively reused test sets (Everingham et al., 2010; Lin et al., 2014), keeps boosting", keeps boosting is a bit of a weird phrase here, "keeps improving" perhaps?

- "suggesting their insufficiency to cover hard examples that may be encountered in the real world", maybe replace "insufficiency" and "cover" with "inability" and "handle".

- "such test sets may only spot an extremely small subset of possible mistakes that the model will make", "spot" is likely the wrong word to use here, maybe contain? But even so, test sets do not contain mistakes, the methods possibly make mistakes on the test set. Consider rewording.

- "The existence of natural adversarial examples (Hendrycks et al., 2019) also
echos such hidden fragility of the classifiers to unseen examples", while I could guess at what the sentence means, it does not really make sense.

- "which possess inherent transferability to falsify different image classifiers with the same type of errors", not sure what you mean by this.

- "It is clear that images in S are visually much harder.", something is missing.

- "weakly labelling method of filtering", what do you mean by this?

- "Specifically, given the target model $f_t$, we let it compete with a group of state-of-the-art segmentation models {g_j}^m_{j=1} by maximizing the discrepancy (Wang et al., 2020) between f_t and g_j on D." They are not really competing are they? The point, as I understand it, is not to select the best model, but to find the most "controversial" image.

- I would have prefered legends in each figure, as opposed to having to scroll up and down to find the relevant information.

- "indicating that many images in $T^{(1)}$ are able to falsify both the target model $f_t$...", the images are not really falsifying the model.

- "This suggests that the target model begins to introspect and learn from its counterexamples", the word introspect appears to be wrongly used here (and elsewhere).

- "Moreover, the top-1 model on $T^{(0)}$ does not necessarily perform the best on $T^{(1)}$ , conforming to the results in (Wang et al., 2020).", what results are you talking about specifically? And I guess it should be "confirming".

Originality
I don't find the work very original and it is not clear to me if the work is very novel. A lot of literature is referenced under related work, but I find it to be mostly tangentially related. It would be good if the authors could describe how this specific problem has been addressed before in image analysis and particularly image segmentation. A google search brings a number of works on hard negative mining. But "human-in-the-loop" techniques such as interactive training [1, 2] also enable annotators to focus more time on harder examples. The methodology itself is not groundbreaking. Multiple trained models have been used in combination to assess prediction certainty previously and uncertainty has also been used in active learning setups to focus annotations on difficult regions [3]. I am sure there are even more relevant links, but this is what a couple of minutes googling brought up.

Significance
While the problem is relevant and the method possibly useful, because of previously mentioned concerns with respect to novelty and generalizability of the results, I do not think it will have a wide ranging significance.

[1] Gonda, Felix, et al. "Icon: An interactive approach to train deep neural networks for segmentation of neuronal structures." 2017 IEEE 14th International Symposium on Biomedical Imaging (ISBI 2017). IEEE, 2017.
[2] Berg, Stuart, et al. "Ilastik: interactive machine learning for (bio) image analysis." Nature Methods (2019): 1-7.
[3] Casanova, Arantxa, et al. "Reinforced active learning for image segmentation." arXiv preprint arXiv:2002.06583 (2020).

---

> ### Author Response · Authors · 2020-11-20
> **Response to Reviewer2 (part 1/3)**
>
> We appreciate the reviewer’s time and helpful feedback. We find the reviewer to be respectably familiar with segmentation and active learning. However, our work comes from a different field that might perhaps be less familiar to the reviewer (assessed from the comments saying our related work section is “tangentially related”), i.e., the work line of Maximum Discrepancy Competition (MAD), which mainly arises from human vision science and scientific philosophy, e.g., (Wang & Simoncelli, 2008; Ma et al., 2018; Wang et al., 2020).  We have clarified point-to-point below, and we sincerely hope the reviewer would take our reply into account and consider a more fair and positive assessment.
>
>
> Q: Overfitting specific or similar-pattern corner cases? Shall we use an independent dataset of corner-cases identified manually?
>
> A: This question seems to misunderstand our whole framework, assumption, and methodology on a fundamental level. We apologize if our presentation has caused any such wrong impression, and re-clarify our main points below.
>
> In each round, the test and training sets are never overlapped. The test MAD images are always re-selected by finding the currently re-trained model’s corner-case samples. In other words, we always adaptively seek the “most falsifying” subset (beyond the training set) from the full unlabeled data, that can degrade the trained model’s performance on it to the most extent.  So there is no overfitting on any specific corner case.
>
> “Corner-cases identified in a similar manner” is also not a correct assessment. We observed that MAD examples found at different rounds display different patterns, as they are changing as we update the target model more, which implies that the original target model could have multiple “weakness” aspects to spot. Intuitively, easy errors are first fixed, and then the next round is still able to dig out more hidden errors. So there is no overfitting on one same corner case pattern, even after multiple rounds.
> Further, we note that the tuned models preserve their generalization on canonical test sets, while keeping improving their robustness remarkably on spotted different catastrophic mistakes after rounds. That is another side evidence that our tuned model is not overfitting a few specific corner cases or similar patterns.
>
> Regarding the independent corner case set selected manually: that would be very ideal to have, but not always feasible. Constructing such a “model-dependent” set requires collecting, annotating, assessing, and selecting hard samples, from a large unlabeled data pool, for every specific model. Obviously, that is significantly beyond the scope and workload of any existing test set, that is model-agnostic and not specialized at spotting corner cases of any specific model.
>
> To review our motivations: (1) we aim to drastically reduce the sample/annotation amount needed while keeping the same effectiveness of probing model generalization; (2) we aim to make the test set adaptive to each specific model (which should differ even just before and after tuning), to avoid “overfitting” a fixed and heavily re-used test set. The two motivations explain why we opt not to do a large, fixed set, and should have also addressed your question too.

---

> ### Author Response · Authors · 2020-11-20
> **Response to Reviewer2 (part 2/3)**
>
> Q: Do segmentation benchmarks require pixel-level dense annotation?
>
> A: This is a good point. We agree segmentation benchmarks could also be built with partial masks or only “key pixel” annotations. We will remove this imprecise statement from the draft (to be updated soon). However, we note that this change has no impact on the whole storyline nor technical merit, as our MAD competition strategy is generally applicable with any annotation granularity.
>
> Q: Testing with imbalanced class.
>
> A: While this statement is NOT the main motivation or storyline of our work (just a mention as background), we share our opinion here: for high-stake applications, we want to avoid sparse but fatal failures. It is quite common that some minority class could be the high-stake one, that we hope to test its segmentation reliability with much more caution. For example, think of “sky” (usually much more pixels per image) versus “pedestrian” (smaller amounts of pixels) classes in an autonomous driving scene: the latter is obviously more critical. That is why we hope to go beyond a fixed test benchmark where some important classes may be underrepresented and not get the thorough testing deserved by their practical high stakes. This can be resolved by exposing to massive unlabeled data and selecting corner cases by MAD.
> We are comfortable to remove this sentence to avoid unnecessary confusions as suggested by the reviewer.
>
> Q: "I find the description of the construction of the test dataset used in the different iterations unclear."
>
> A: To probe the generalization to the real world, especially the corner-case performance is a wide-open problem in image segmentation. As a result, it is highly nontrivial to manually construct an “independent” dataset that contains sufficiently representative hard and corner-case examples. Our method makes one of the first attempts to efficiently expose failures of image segmentation models in the real world, and to leverage such failures for improved generalizability without degrading performance on standard test sets. We agree that a measure of representativeness of the selected hard samples in T(1), T(2), and T(3) is worth in-depth investigation. But we also would like to point out T(1), T(2), and T(3) are selected independently, on which the performance provides a reasonable estimate of the generalizability of fine-tuned models to the real world.
>
>
>
>
> Q: Disagree with the statement “This also provides direct evidence that existing segmentation models could be particularly weak at certain real-world generalization...”
>
> A: Here the generalization ability mainly refers to the performance on real-world hard corner case samples. It is also “evidence that the selection procedure is working as intended”. We will revise the sentence in the main text to:
> "This also provides direct evidence that hard corner cases of existing segmentation models could be exposed. It is also proof-of-concept that the selection procedure is working as intended.”

---

> ### Author Response · Authors · 2020-11-20
> **Response to Reviewer2 (part 3/3)**
>
> Q: Novelty. Relation with active learning and interactive learning.
>
> First, our work is derived mainly from the classical field of MAD, a perceptually and psychologically grounded framework to efficiently spot human or model failures. Starting from the foundational framework, and introduced by the philosopher of science Karl Popper, falsifiability sets the limits of scientific theories. The idea of model falsification as model comparison has been successfully applied to the field of vision science to compare perceptual discriminable quantities such as visual quality and visual aesthetics. we quote R3’s comments who have very precisely identified our most related prior work, and our innovations on top of that one:
>
> “This work extends from MAD, ICLR 2020 (https://openreview.net/forum?id=rJehNT4YPr); but it also presents with two nontrivial and interesting innovations: (1) generalizing it to a dense prediction task, which requires revising the human labeling strategy in subjective experiments. Weakly-supervised labeling is more practical for segmentation; and (2) extending to active training/tuning, leveraging the selected hard examples to improve the segmentation model for multiple rounds. The tuned models were shown to improve their robustness remarkably on spotted catastrophic mistakes, while preserving their performance on canonical testing sets.”
>
> We accept your suggestion that our method might also be linked to active learning and interactive/human-in-the-loop training. The authors are no strangers to active learning either; although we consider our framework to be brewed from a different stream of philosophy from active learning. If taking the alternative lens of active learning to view our work, we are introducing a new active learning criterion that the selected images have the greatest potential to make the target model “disagree” with another auxiliary competitive model. In fact, we have compared with the classical entropy-based active learning in Appendix A, whose criterion is to maximize the prediction uncertainty of the target model. As described in Appendix A, all experimental settings (including target model structure, human labelling budget, etc.) are kept identical. The only difference comes from the sampling strategy. Results show that the model achieved by our method is winning with a noticeable mIoU advantage, clearly suggesting its superior performance on open-world images than the model fine-tuned by the entropy-based method.
>
> All other suggested typos and sentence revisions will be (in a short time window) addressed in the revised paper. Thanks for your careful reading!
>
> We sincerely hope after the above clarification, the context and originality of our work have become clear to R2.

---

### Official Review · AnonReviewer3 · 2020-10-29
**Interesting Premise, Could Use Further Refinement**

**Rating:** 4
**Confidence:** 4

**Review:**

Summary:
In this work, the authors seek to leverage external sources of data to improve the generalization of segmentation models. In particular, they seek to identify images which generate discordance among models, hypothesizing that they would be well-suited to improve model performance. Once selected, they leverage human annotators to first filter this image set and then segment the images, which are then used to retrain the model. They demonstrate improved performance relative to a batch of competing models which are not updated using this procedure.

------

Recommendation:
Given the lack of a competitive baseline, opaqueness around the impact of the algorithm's hyperparameters, and what's likely to be noisy estimates of improvement, I cannot recommend this paper for acceptance as is. See below for greater detail.

------

Positives:
* The authors recognize that not all images are created equal when looking to improve a model and attempt to tackle this challenging problem. This is particularly relevant in high-stakes environments when failure in rare cases can have a disproportionate impact (e.g., autonomous vehicles, healthcare, etc.).

* The authors identify that scaling human annotation, in particular for image segmentation, can be cost prohibitive and propose a method to optimize this process. If successful, such a method could have significant implications for industries where the cost of annotation is high (e.g., healthcare where highly paid experts are required).

---

Concerns:
* The authors propose a fairly complex (and costly) pipeline for improving generalization to the unseen dataset. However, they fail to compare against even a simple baseline such as random selection of images from this dataset. Comparing against models which are not updated, in particular when it's clear that none of them generalize, is a weak baseline that could likely be outperformed by far simpler uses of the external data.

* T^(i+1) = 30 is quite small, especially given the number of classes. While it is understandable that such a sample cannot be extraordinarily large, by leveraging such a small value there's significant noise in the evaluation criteria. Given the previous concern, this likely would not affect the reported results in the paper. However, with a more competitive baseline, such noise may make it challenging to identify improvements in the selection of image for finetuning.

* There are a few magic constants throughout the paper. While conducting an ablation study may be cost-prohibitive given the sequential dependencies of the algorithm, it would be helpful to the reader to provide some means of estimating the impact of and sensitivity to these parameters.

---

Nits:
* The comment on maximum test set size on page 1 is a bit strong. Test set sizes are limited by financial incentives. For potentially lucrative endeavors, it would not be surprising to find a test set larger than 10k images.

* There is a typo on page 2: "not be[en] spotted beforehand"

* It would be nice to quantify the computational cost of constructing M in each iteration.

* Consider moving the superscript 4 after the period to make it clear that it is a footnote and not exponentiation.

* There is a typo on page 4:(ARC) -> (ACR)

* The F in "failure" is erroneously capitalized in the first paragraph of section 3.2

* You may wish to consider citing the field of computer-assisted annotation as relevant work. One such paper (among others) and open-source implementation include:
** Efficient Interactive Annotation of Segmentation Datasets with Polygon-RNN++ (Acuna et al)
** https://developer.nvidia.com/blog/annotation-transfer-learning-clara-train/

---

> ### Author Response · Authors · 2020-11-20
> **Response to Reviewer3 (part 1/2)**
>
> We appreciate the reviewer’s time and helpful feedback. We however would like to respectively point that, that current reviews seem to misunderstand several points in our experiment and methodology strength. We have clarified point-to-point below, and we sincerely hope the reviewer would take our reply into account and consider a more fair and positive assessment.
>
>
> Q: Not comparing with other simpler sampling such as random sampling
>
> A: We compared our method with the entropy-based sampling method in Appendix A, which has been shown to be (much) more competitive than random sampling. Specifically, we conducted a MAD competition on the two models obtained from our method and entropy-based method. Results show that the model achieved by our method is winning the competition with a noticeable mIoU advantage, very clearly suggesting its superior performance on open-world images than the model fine-tuned by the entropy-based method.  We are happy to include another random sampling experiment if further suggested to do so, although we sincerely do not feel it necessary based on the above explanation.
>
>
> Q: Testing set size is too small
>
> A: We see a very clear misunderstanding here, of our different-from-usual definition on our “test set”. Our methodology follows the well-established MAD competition philosophy: it aims to select the most challenging small subset of corner-case examples, that are best at falsifying the given model. In other words, the test set has been adaptively crafted to the “worst-case” samples w.r.t. the model, and therefore its requirement on test set size to yield stable, consistent performance evaluation is drastically different from the typical test set consisting of i.i.d. selected, “average-case” examples. This is actually the fundamental motivation underlying our paper, e.g., one could leverage such a small set of “worst-case” examples to trim down the needed annotations while still reliably probing the model’s real-world generalization.
>
> The past seminal literature in MAD, e.g., (Wang & Simoncelli, 2008) and (Wang et al., 2020), have already made several solid arguments on how resilient the MAD evaluation is to the test sample size. For example, please refer to Figure 5 in (Wang et al., 2020), which demonstrates the classification model’s generalization ranking probed by MAD will remain stable and consistent, with as small as 15 images selected for MAD annotation and testing. Following their same idea, we tried to enlarge our test size from 30 to 40, and showed that our MAD competition results remain stable as we increase the testing set size: see Appendix C in the updated paper.
>
> Therefore, under our framework, “leveraging such a small value there's significant noise in the evaluation criteria” and “such noise may make it challenging to identify improvements in the selection of image for finetuning” do not constitute any real concern. We hope the above clarification has cleaned the water.
>
>
> Q: “A few magic constants”
>
> A: We agree with the reviewer that some hyperparameters shall be discussed in more detail. Mostly, they are chosen to be small due to time/human/finance resource constraints, but using the current setting we already observe the substantial improvements after fine-tuning, which proves our concept and that our framework can be very effective even under a tight budget constraint.  We will explain below and will include those discussions in the paper with a separate paragraph at the end of Section 4.
>
> 1. Larger n4 (number of the labeled hard samples by humans) and r (number of finetuning rounds) are positively correlated with the achievable model generalization ability.
>
> 2. Based on the assumption that different models can spot different aspects of failures of the target model, larger m (number of competing models) is supposed to improve troubleshooting too, but also subject to higher constraints.
>
> 3. The number n3 controls the tradeoff between the human effort of weakly labeling and the content diversity of set M. A larger n3 means more candidate hard samples selected for the weakly labeling process, which will lead to more diversity in the selected hard samples.

---

> ### Author Response · Authors · 2020-11-20
> **Response to Reviewer3 (part 2/2): Nits**
>
> Q: The comment on the maximum test set size on page 1 is a bit strong.
>
> A: We agree that “for potentially lucrative endeavors, it would not be surprising to find a test set larger than 10k images”, and will tone down this sentence. But we also stress that given the enormous amounts and dimensions of image data, even 10k images are still deemed to be extremely sparsely distributed in the natural image manifold. Another problem with the conventional model comparison methodology is that (quoting (Wang et al., 2020): “the test sets are pre-selected and therefore fixed. This leaves the door open for adapting classifiers to the test images, deliberately or unintentionally, via extensive hyperparameter tuning, raising the risk of overfitting. As a result, it is never guaranteed that image classifiers with highly competitive performance on such small and fixed test sets can generalize to real-world natural images with much richer content variations.” Therefore, even larger commercial sets are possible (though may not be affordable or sufficiently accessible to general users), our efforts to identify small, adaptive, and worst-case testing sets are still necessary and complementary.
>
>
> Q: Cost of constructing M.
>
> A: Since the images in M are selected in order of their corresponding distance values (Algo 1, line 9), the time cost is mainly determined by the size of M. Selecting n images to M costs approximately 1/10 time compared with seeking dense segmentation maps on n images.
>
>
> Q: Citing the field of computer-assisted annotation as relevant work.
>
> A: Thanks for bringing “Efficient Interactive Annotation of Segmentation Datasets with Polygon-RNN++” to our attention. We will cite it and look at it more closely in future experiments. All other suggested typos and missing references will also be addressed in the paper (to be updated very soon). Thanks again for your careful reading!

---

### Decision · Program_Chairs · 2021-01-07
**Final Decision**

**Decision:**

Reject

**Comment:**

This paper studies how to efficiently expose failures of "top-performing" segmentation models in the real world and how to leverage such counterexamples to rectify the models. The key idea is to discover most "controversial" samples from massive online unlabeled images. The approach is sound, well grounded, and quite logical. Results demonstrate the effectiveness.

However, there exists some limitations coming from R2 and R3, for example, 1) Segmentation benchmarks may not require pixel-level dense annotation. There are also examples of benchmarks where the groundtruth consists of computer segmentations corrected by humans. 2) It is much harder for segmentation data to be class-balanced in the pixel level, making highly skewed class distributions common for this particular task. 3) Citing the field of computer-assisted annotation as relevant work.

In the end, I think that this paper may not be ready for publication at ICLR, but the next version must be a strong paper if above limitations can be well addressed.